# Failure Mechanism of Tensile CFRP Composite Plates with Variable Hole Diameter

**DOI:** 10.3390/ma16134714

**Published:** 2023-06-29

**Authors:** Pawel Wysmulski

**Affiliations:** Department of Machine Design and Mechatronics, Faculty of Mechanical Engineering, Lublin University of Technology, Nadbystrzycka 36, 20-618 Lublin, Poland; p.wysmulski@pollub.pl

**Keywords:** CFRP composite, damage mechanics, crack propagation, tensile analysis, finite element method

## Abstract

Real thin-walled composite structures such as aircraft or automotive structures are exposed to the development of various types of damage during operation. The effect of circular hole size on the strength of a thin-walled plate made of carbon fibre-reinforced polymer (CFRP) was investigated in this study. The test object was subjected to tensile testing to investigate the strength and cracking mechanism of the composite structure with variable diameter of the central hole. The study was performed using two independent test methods: experimental and numerical. With increasing diameter of the central hole, significant weakening of the composite plate was observed. The study showed qualitative and quantitative agreement between the experimental and numerical results. The results confirmed the agreement of the proposed FEM model with the experimental test. The novelty of this study is the use of the popular XFEM technique to describe the influence of the hole size on the cracking and failure of the composite structure. In addition, the study proposes a new method for determining the experimental and numerical damage and failure loads of a composite plate under tension.

## 1. Introduction

The design of modern structures with optimised strength and stiffness parameters requires the use of advanced technologies. This particularly applies to high-tech aerospace or automotive structures in which the most beneficial solutions in terms of operation and durability can be obtained by e.g., replacing previously used materials with advanced composite materials [1,2,3]. These primarily include polymer laminates reinforced with continuous fibres, predominantly carbon fibre-reinforced plastics (CFRPs) and glass fibre-reinforced plastics (GFRPs). Due to very favourable mechanical properties of these materials in relation to their own weight, it has become possible to use fibre composites in the production of carrying elements in thin-walled structures (e.g., for covering reinforcement profiles) [4,5,6,7,8]. Laminates make it possible to shape the mechanical properties of designed components in terms of their ability to carry the desired type of load. As a result, it is possible to achieve very advantageous construction designs; this, however, requires the use of modern testing methods that enable the structural performance to be analysed over the full range of loading conditions [9,10,11,12]. Previous studies on composite structures reported in the literature mostly focus on analytical and numerical considerations, with analyses conducted on structures with typical cross-sections, operating under ideal conditions and subjected to simple loading cases: compression, shear, or simple bending. Only to a limited extent are such considerations verified by experimental tests on real construction elements [13,14,15].

In layered composites (laminates), the state of stress is a complex issue because it depends on the fibre configuration and varies from layer to layer [16]. Therefore, the stresses induced by a hole in the laminate vary from layer to layer, and the classical Kirsch problem for isotropic materials cannot be applied in such cases [17]. The currently popular numerical methods could help in such cases [18,19]. As is known from scientific publications, the occurrence of holes in thin-walled structures is unavoidable, if only for technological reasons [20,21,22]. The complexity of the above issue, resulting from the possibility of designing the material properties of laminated composites, makes this topic still valid for researchers. The literature [23,24,25,26] offers solutions to the problem of hole formation in composite materials.

The extended finite element method (XFEM) eliminates the need for a conformal finite element mesh [27]. The extended finite element method was first introduced by Belytschko and Black [28]. It is an extension of the conventional finite element method based on Melenko and Babuska’s concept of partition of unity [29], which allows local enrichment functions to be easily incorporated into the finite element approximation. The presence of discontinuities is provided by special enrichment functions in combination with additional degrees of freedom. However, the finite element structure and its properties, such as sparsity and symmetry, are retained. The use of the XFEM method makes it possible to study crack initiation and propagation along any path without numerical model remeshing [27]. Moving cracks are modelled using one of two alternative approaches: the cohesive segment approach or the linear elastic fracture mechanics (LEFM) approach [27,30,31]. Using these techniques, crack initiation is defined up to the onset of cohesive degradation in the enriched component, and the degradation stage occurs when the stresses or strains meet the specified crack initiation criteria [32].

The present study analysed the effect of variable diameter of the central hole on the behaviour of a thin-walled composite plate under tension. The study was carried out on plates weakened by a central hole with diameters of 2 mm, 4 mm, and 8 mm. The tensile tests were carried out over the full range of loading, from failure initiation through crack propagation to complete failure of the composite structure [33,34,35,36]. The research was carried out using two independent methods: experimental and numerical using FEM. This approach made it possible to develop numerical models that closely reproduced real plates [37,38,39,40].

The novelty of the research problem undertaken in this study is that it describes the fracture and failure of a hole-weakened composite plate using the currently popular XFEM technique. The effect of hole size on the strength of the composite plate is investigated. The state-of-the-art Aramis measuring system is used in experiments. In addition, the study proposes a new method for determining the crack initiation and failure loads of the composite plate under tension, and the results obtained thereby are verified numerically. A review of the literature shows that many studies have used XFEM for isotropic materials [41,42,43,44]. However, there is a lack of studies describing the failure of real sandwich composites (CFRPs) using the numerical XFEM technique. A numerical model created based on experimental results can be used to analyse the failure process for such composite elements.

## 2. Object and Methodology

The test object was a thin-walled laminated composite plate. Holes were cut in the plate with diameters of 2 mm, 4 mm, and 8 mm. The holes were used to weaken the structure and to cause the composite to crack in a specified area during the tensile test. The test object was made from a unidirectional HeXPly prepreg strip (from Hexcel) of carbon fibre-reinforced composite in an epoxy matrix. The polymerisation process took place in an autoclave. The curing process was carried out in a package vacuum of 0.08 MPa, overpressure of 0.4 MPa, and autoclave temperature of 135 °C for 2 h. The laminate structure had a symmetrical fibre arrangement of the composite layers [0/90/0/90_2_/0/90/0]—Figure 1b.

Table 1 shows the mechanical properties of a single layer of CFRP laminate in three orthotropic directions. The properties of the carbon–epoxy composite were determined experimentally in compliance with the ISO standard and as described in [45]. This allowed for obtaining real mechanical properties of the produced material, as they may differ from the ideal properties specified by the manufacturer. The determined properties were used to define the material model in a numerical analysis conducted by the finite element method using Abaqus.

Figure 1a shows the test object, which consisted of 3 plates with dimensions of 16 mm (width) × 180 mm (length) × 1.048 mm (overall thickness). For each specimen, an oval hole with a diameter of H = 2 mm (Figure 1c,d), H = 4 mm (Figure 1e,f), and H = 8 mm (Figure 1g,h) was made in the centre of the plate. The specimens were painted in contrasting patterns (Figure 1d,f,h). The samples were measured during the experiments using a non-contact optical measuring method.

The ARAMIS optical measuring system is designed for non-contact displacement measurements in planar and spatial components under load. It consists of a set of cameras recording changes in the shape of the object under test and a suitably adapted and programmed computer storing and processing the recorded images. Depending on the configuration, i.e., the number and speed of cameras, the system can be used to analyse displacement and deformation fields of flat or spatial elements under static or dynamic loading.

The measurement principle is the same as in photogrammetry, i.e., on the basis of the images, the spatial coordinates of selected points are determined. Measurement proceeds as follows. A photo of the object in its undeformed state is taken, followed by a series of photos corresponding to the successive loading stages of the object. Each of the photographs is then compared with the output and a set of displacement values of selected points on the surface of the object is created. The selected points are points of interest on the surface of the object. They can be things such as spots, dots, or other colour changes naturally occurring on the surface. If the surface is low-contrast without visible colour changes, it is first painted with white paint with sufficient adhesion and then tinted, preferably with black spray paint to create an irregular pattern. Using this irregular pattern, the area analysis programme creates a grid of analysed points. These points are the centres of so-called “facets”, i.e., the centres of small areas into which the entire analysed area has been divided. The programme records the coordinates of these points, then determines changes in their position and further determines deformations on this basis. The program records the coordinates of these points and determines the changes in their position and, on this basis, determines the deformations, logarithmic or Green’s.

The manufactured plates with a central oval hole were subjected to axial tensile testing. The Cometech QC-505 M2F (Taichung City, Tajwan) universal testing machine (Figure 2 item 1) equipped with a load cell with a range up to 50 kN and an accuracy class of 0.5% (Figure 2 item 3) was used in the experiments. Specially designed wedge grips with facings for flat specimens with a thickness ranging from 0.2 to 11 mm were attached to the pivots of the measuring machine (Figure 2 item 4). They were used to constrain the test specimen (Figure 2 item 5), which was inserted axially, 30 mm each into the upper and lower grips. This made it possible to obtain a 16 × 120 mm test area of the plate. During the experimental tensile test, the load and elongation of the plate hole were measured with a constant upper crosshead speed (Figure 2 item 2) of 1 mm/min. In addition, the displacement of the composite structure in the frontal plane of the plate over time was recorded using the Aramis non-contact optical measuring system (Figure 2 item 6). This system is equipped with a 20 M resolution camera (5472 × 3648 px) and has a working area from 20 × 15 mm^2^ to 5000 × 4000 mm^2^, which allows for the sample to be observed with images captured at up to 17 Hz. The use of this system made it possible to examine the behaviour of the plate during the tensile test, causing it to crack and fail. The experiments were conducted in accordance with the ASTM D5766 Standard Test Method for Open-Hole Tensile Strength of Polymer Matrix Composite Laminates [46].

The numerical analysis was performed by the finite element method using the commercial version of the Abaqus system [47]. An adequate FEM model corresponding to the experimental sample was prepared. To that end, a CAD model of a 16 × 120 object with a thickness of 0.131 mm was designed, with oval holes made at the centre point of the plate with diameters of 2 mm, 4 mm, and 8 mm. The mechanical properties of the material of the numerical model were assigned in accordance with Table 1. The structure of the laminate was made by modelling the layers as separate solids. The FEM model consisted of 8 solids of 16 mm (width) × 120 mm (length) × 0.131 mm (thickness) stacked layers. For each solid, the fibre stacking orientation was assigned according to the laminate configuration [0/90/0/90/90/0/90/0]. In order to speed up the FEM computational time, the numerical model was limited to the test area of the specimen used in the experiment (16 mm × 120 mm × 1.048 mm). Therefore, boundary conditions were defined for the cross-sections of the FEM model. The lower cross-section was fully restrained by taking away all degrees of freedom as Ux = Uy = Uz = URx = URy = URz = 0, while the upper cross-section was blocked with two translational degrees of freedom as Ux = Uz = 0 and with three rotational degrees of freedom as URx = URy = URz = 0. To apply tension, the upper cross-section was assigned a displacement of 5 mm, as shown in Figure 3a. In addition, the longitudinal edges of the plate were defined as Uz = 0. In this multilayer FE model, each layer was solid and treated as a unidirectional continuous layer [48]. These orthotropic laminae were connected through Tie relations to form a lamina, thus creating perfectly connected but discontinuous interfaces between the laminae. Furthermore, this multilayer FEM model allowed the behaviour of the lamina interfaces to be made explicit in order to simulate the damage and failure process of the FRP composite laminate. When configuring the XFEM analysis, the contact interaction property was selected for each layer of the laminate in order to define the tensile crack surface behaviour. The interactions of the FEM model are represented schematically in Figure 3b. The numerical model was discretised with hexahedral solid elements of C3D8R type, having linear interpolation with 8 nodes and reduced integration. For structural meshing, partitions of the FEM model were made and the finite element mesh density was increased around the circumference of the circular hole, as shown in Figure 3c,d. The finite element size was adopted based on a preliminary numerical analysis, which proved that reducing the finite element size did not affect the quality of the numerical results. Reducing the size of the elements near the holes did not affect the results, but largely increased the CPU time.

### Theory of XFEM

The use of the extended finite element method (XFEM) allows the study of crack initiation and propagation without the need to re-mesh the model [27]. For crack analysis, enrichment functions typically consist of asymptotic near-tip functions that capture the singularity around the crack tip and a discontinuous function that represents the displacement spike on the crack surfaces. The nodal displacement vector enrichment function *u* is expressed as [49,50]
(1)$$u=\sum_{I=1}^{N}N_{I}\left(X\right)\left[u_{I}+H\left(X\right)A_{I}+\sum_{\alpha =1}^{4}F_{\alpha }\left(X\right)B_{I}^{\alpha }\right]$$
where $N_{I}\left(X\right)$ is the nodal shape function, $u_{I}$ is the nodal displacement vector of the continuous part of the finite element solution, $H\left(X\right)$ is the discontinuous jump function across the crack surface; $A_{I}$ is the nodal vector of degrees of freedom; $F_{\alpha }\left(X\right)$ is the elastic asymptotic crack tip function; $B_{I}^{\alpha }$ is the nodal vector of degrees of freedom. While the first segment of the formula applies to all nodes in the model, the second segment is valid for the nodes whose shape function support is intersected by the crack interior, and the third segment is only used for the nodes whose shape function support is intersected by the crack tip. Figure 4 illustrates the tangential and normal directions (with respect to the crack) at various points along the crack interior as well as the crack apex. It also illustrates the local polar coordinate system at the crack tip.

Asymptotic singularity functions are only taken into account when modelling stationary cracks in Abaqus/Standard. Within the moving cracks $F_{\alpha }\left(X\right)=0$ and the nodal enrichment function, the displacement vector u is as follows:(2)$$u=\sum_{I=1}^{N}N_{I}\left(X\right)\left[u_{I}+H\left(X\right)A_{I}\right]$$

The discontinuous jump function across the fracture surface $H\left(X\right)$ could be represented as follows [50]:(3)$$H\left(X\right)=\left\{\begin{array}{c} 1\;if\;\left(X-X^{*}\right)\cdot \;n\geq 0\\ -1\;otherwise\;\;\;\;\;\;\;\;\;\;\;\;\;\;\;\;\; \end{array}\right.$$
where X is the sample point (Gauss), $X^{*}$ is the point on the crack closest to X, n the unit outward normal to the crack at $X^{*}$.

Moving cracks are modelled in Abaqus using one of two alternative approaches: the cohesive segment approach or the linear elastic fracture mechanics (LEFM) approach [27]. Crack initiation is defined to the onset of cohesive degradation in the enriched component. In contrast, the degradation stage occurs when the stresses or strains meet certain crack initiation criteria. One of these criteria is the maximum principal stress criterion (MAXPS), which is expressed as follows:(4)$$F=\frac{\langle \sigma_{MAX}\rangle }{\sigma_{MAX}^{0}}$$
where $\sigma_{MAX}^{0}$ is the maximum permissible principal stress.

The maximum principal stress ratio $\langle \sigma_{MAX}\rangle$ shown in the Macaulay brackets assumes that the damage begins when the value equals 1:(5)$$\left\{\begin{array}{c} \langle \sigma_{MAX}\rangle =0\;\;\;\mathrm{if}\;\;\;\sigma_{MAX}<0\\ \langle \sigma_{MAX}\rangle =\sigma_{MAX}\;\;\;\mathrm{if}\;\;\;\sigma_{MAX}\geq 0 \end{array}\right.$$

## 3. Results and Discussion

An analysis of the effect of central hole diameter on the strength of the tensile plate was performed in four stages. The process of plate deformation, failure initiation, crack propagation, and failure of the CFRP composite material was described. The study proposes a new method for determining the experimental and numerical damage and failure loads of a composite plate under tension. The analysis was carried out using two independent methods simultaneously: experimental and numerical.

### 3.1. Plate Deformation

The experimental tensile testing of a composite plate with variable-diameter hole was conducted using the Aramis non-contact optical measuring system to measure the displacement of the specimen during the whole test. In addition, this measuring system made it possible to generate graphical displacement maps superimposed on real plates. Figure 5 shows the elongation analysis results for the plates with 2 mm, 4 mm, and 8 mm diameter holes. The proposed experimental method allowed the elongation of the specimens to be measured before complete failure. The highest elongation of 1.175 mm was obtained for the plate with a central hole diameter of 2 mm, which accounted for 1.1% of the elongation of its length (Figure 5a–c). On the other hand, the lowest elongation of 0.91 mm (0.6% elongation) was obtained for the plate with an 8 mm diameter hole (Figure 5h–j). In addition, the elongation of the hole was measured using the Aramis system, yielding an elongation of 8.8% for the plate with a 2 mm diameter hole, 6.6% for the plate with a 4 mm diameter hole, and 4% for the plate with an 8 mm diameter hole. The experimental findings showed that increasing the diameter of the hole resulted in a decrease in the total elongation of the plate. The experimental results were then compared with the results of the numerical analysis. The deformations obtained with both test methods used were found to be consistent. The maximum numerical elongation before failure occurred for the same plate and was 1.175 mm (Figure 5c), while the minimum occurred for the 8 mm diameter hole and amounted to 0.92 mm, which agreed with the experimental measurements. An analysis of the results showed qualitative and quantitative agreement between the experimental and numerical findings. The results confirmed the agreement of the proposed FEM model with the experimental test.

### 3.2. Crack Initiation

Figure 6 presents the onset of the cracking process in the laminate structure for all tested plates. The numerical cracking process was determined by XFEM. The cracking started when a value of 1 was reached according to the maximum principal stress ratio criterion. For all tested plates, the damage of the composite structure initiated with transverse cracking of the outer layer with a 0° fibre orientation in the area of the circular hole, as shown in Figure 6a–c. It should be added that the damage criterion initiated cracking of other laminate layers with a 0° fibre orientation.

### 3.3. Failure of the Composite

The experimental and numerical investigation was carried out over the full range of tensile loading until complete failure of the composite. Figure 7a,c,e show the experimental failure mode of the analysed plates with variable-diameter holes. The size of the hole did not affect the mode of cracking; for all cases, the real specimen cracked in the expected area where it had previously been weakened by the hole. The crack path passed across the plate halfway along its length. The parallel numerical analysis showed the same failure mode, as presented in Figure 7b,d,f. The crack propagation in the FEM model (Figure 6) initiated in the same area as that observed in the experiment (Figure 7a,c,e) and proceeded in the transverse direction. The strength of the plate with holes depended on the strength of the layers with a 0° fibre orientation, which were the most stressed in the tensile test. 

### 3.4. Damage and Failure Loads

Tensile load as a function of specimen elongation was measured experimentally. The experiments were extended to include a plate without a hole. This made it possible to determine the working paths of the tensile plates over the full range of loading until failure. The same characteristics were determined for the numerical model. This allowed validation of the experimental and numerical working paths, which are summarised in Figure 8a–d. For all cases, the numerical working path is stiffer than the experimental one, which is due to the fact that the numerical model was not exposed to material imperfections that may occur in the real plate. This approach allowed the damage load corresponding to the initiation of laminate cracking and the failure load causing complete failure of the composite structure to be determined graphically for all cases under study. The damage load P_d(EXP)_ corresponded to the first sudden increase in elongation measured along the working path, while the failure load P_f(EXP)_ was determined at the point of sudden decrease in the tensile load (Figure 8a–d). In the numerical analysis, P_d(FEM)_ and P_f(FEM)_ were determined in the same way as in the experiment. 

Table 2 presents the experimental and numerical damage and failure load values measured for the plate without a hole and for the plate with a hole with diameters 2 mm, 4 mm, and 8 mm. For all cases, the numerical damage initiation load P_d(FEM)_ and the failure load corresponding to complete failure of the composite due to cracking P_f(FEM)_ were higher than the corresponding experimental loads P_d(EXP)_ and P_f(EXP)_. The highest stiffness was obtained for the plate without a hole, for which P_d(EXP)_ was 9031 N and P_f(EXP)_ = 13,484 N. In contrast, the lowest stiffness was obtained for the plate with an 8 mm diameter hole, for which P_d(EXP)_ = 2393 N and P_f(EXP)_ = 6299 N. The error in prediction between the numerical and experimental values of the P_d_ load was in the range of <7% ÷ 15%>. The error in prediction between the experimental and numerical failure load was in the range of <6% ÷ 14%>. Based on the results, the percentage increase in the composite structure failure load P_f(EXP)_ relative to the failure initiating load P_d(EXP)_ was determined. It was found that after reaching the damage load value, the tensile real structure could still carry a load increased by <49% ÷ 163%>. The largest increase in the failure load relative to the damage load was obtained for the plate with an 8 mm diameter hole and the lowest for the plate without a hole. 

In order to demonstrate the influence of the hole on the tensile behaviour of the composite plate, the experimental and numerical working paths for all tests are compared in Figure 9. The experimental and numerical working paths show the expected agreement between the results. The experimental paths reveal a decrease in the stiffness of the plate with increasing hole diameter. The highest decrease in stiffness was observed for the plate with an 8 mm diameter hole. For this case, the damage load P_d(EXP)_ (initiating cracking) decreased by 73% and the failure load P_f(EXP)_ by 53% compared to the plate without a hole. An analysis of the numerical working paths showed a similar effect of hole size on the decrease in plate stiffness—Figure 9. The maximum decrease in the numerical damage load P_d(FEM)_ was 72%, and the maximum decrease in the cracking load P_f(FEM)_ was 57% and was obtained for the plate with the largest hole diameter. The results showed qualitative and quantitative agreement between the experiment and the numerical analysis. The results also confirm the relevance of the developed numerical FEM/XFEM model to the experiment.

## 4. Conclusions

The proposed experimental method made it possible to measure the elongation of the specimens before they underwent complete failure. The highest elongation of 1.175 mm was obtained for the plate with a 2 mm diameter central hole, which accounted for 1.1% of its elongation. On the other hand, the lowest elongation of 0.91 mm (0.6% of the specimen elongation) was obtained for the plate with an 8 mm diameter hole. In addition, the Aramis system was used to measure hole elongation, yielding an elongation of 8.8% for the plate with a 2 mm diameter hole, 6.6% for the plate with a 4 mm diameter hole, and 4% for the plate with an 8 mm diameter hole. The experimental findings showed that increasing the diameter of the hole resulted in a decrease in the total elongation of the plate.

The size of the hole did not affect the mode of cracking; for all cases, the real specimen cracked in the expected area where it had previously been weakened by the hole. The crack path passed across the plate halfway along its length. The strength of the plate with 2 mm, 4 mm, and 8 mm diameter holes depended on the strength of the layers with a 0° fibre orientation, which were the most stressed in the tensile test.

The study also proposed a new method for determining the experimental and numerical damage and failure loads of a composite plate under tension. For all cases, the numerical value of the damage initiation load P_d(FEM)_ and the failure load corresponding to the total failure of the composite due to cracking P_f(FEM)_ were higher than the corresponding experimental loads P_d(EXP)_ and P_f(EXP)_. The highest stiffness was obtained for the plate without a hole, for which P_d(EXP)_ was 9031 N and P_f(EXP)_ = 13,484 N. In contrast, the lowest stiffness was obtained for the plate with an 8 mm diameter hole, for which P_d(EXP)_ = 2393 N and P_f(EXP)_ = 6299 N. The error in prediction of the numerical and experimental values of the P_d_ load was in the range of <7% ÷ 15%>. The error in prediction of the experimental and numerical failure load was in the range of <6% ÷ 14%>. The tensile real structure was still able to carry a load increased by <49% ÷ 163%> after reaching the damage load value. At the same time, the largest increase in the failure load with respect to the damage load was recorded for the plate with an 8 mm diameter hole, while the smallest was for the plate without a hole. 

The largest decrease in stiffness was observed for the plate with an 8 mm diameter hole. For this case, the damage load P_d(EXP)_ (initiating cracking) decreased by 73% and the failure load P_f(EXP)_ by 53% compared to the plate without a hole. In contrast, the maximum decrease in the numerical damage load P_d(FEM)_ was 72%, and in the failure load P_f(FEM)_, it was 57% and occurred for the plate with the largest hole diameter. The results showed qualitative and quantitative agreement between the experiment and the numerical analysis. The results also confirmed the adequacy of the developed numerical FEM/XFEM model to the experiment.

Future research will investigate the influence of composite material layer configuration and the number of layers on crack propagation. As part of the research, experiments will be performed and their results will be validated numerically using the popular finite element method (FEM). The research methodology and conclusions described in this paper will be used in the future study.

## Figures and Tables

**Figure 1 materials-16-04714-f001:**
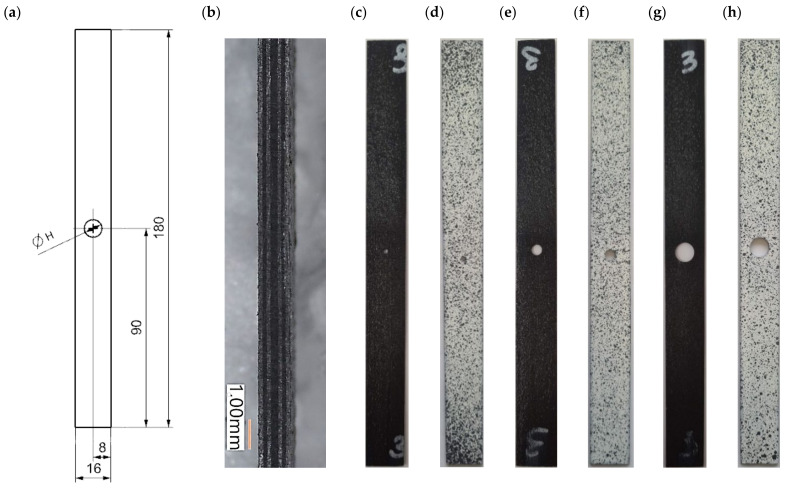
Real composite plate with a drilled hole: (**a**) sketch with dimensions, (**b**) section layup, (**c**) hole 2 mm, (**e**) hole 4 mm, (**g**) hole 8 mm, (**d**,**f**,**h**) specimen with contrasting pattern.

**Figure 2 materials-16-04714-f002:**
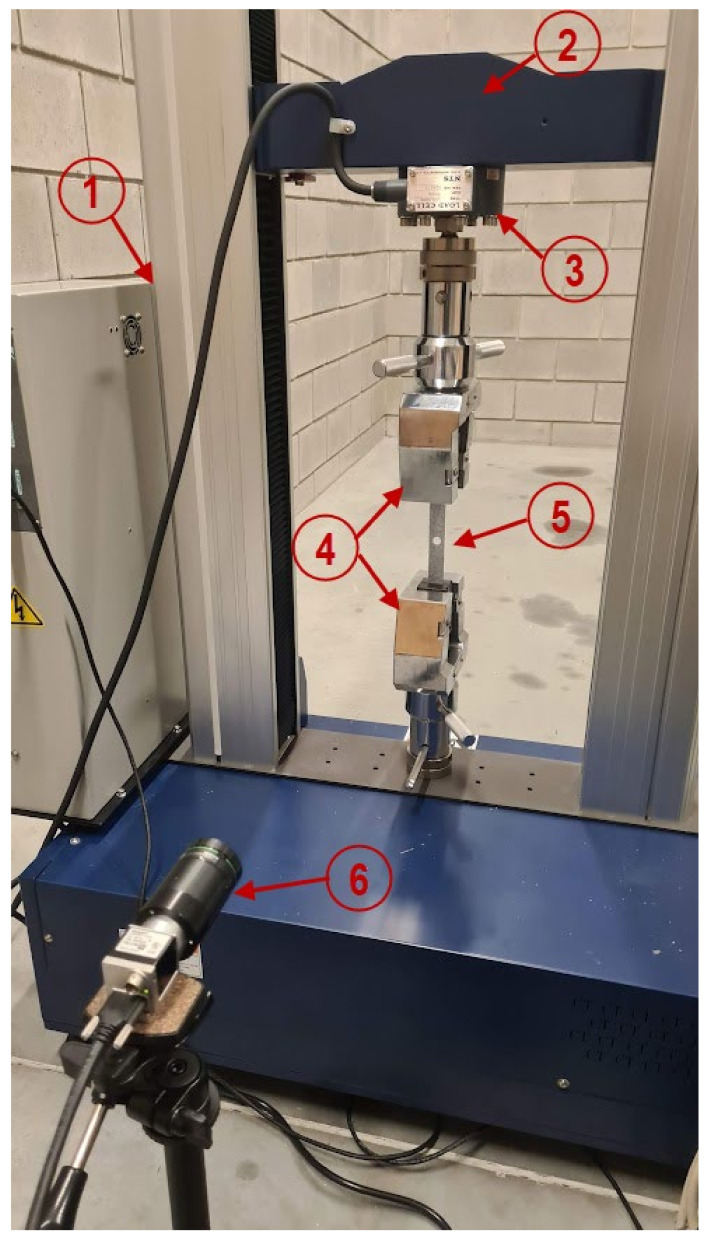
Experimental test stand: 1—testing machine, 2—upper crosshead, 3—measuring head, 4—wedge grips, 5—test specimen, 6—Aramis system.

**Figure 3 materials-16-04714-f003:**
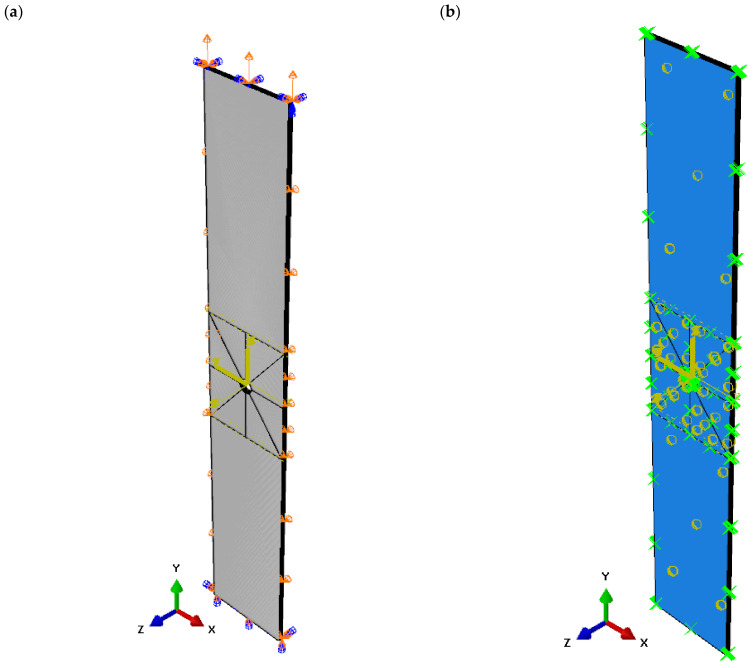
Numerical model: (**a**) implementation of boundary conditions, (**b**) interlayer interactions, (**c**) discretisation hole 2 mm, (**d**) discretisation hole 8 mm.

**Figure 4 materials-16-04714-f004:**
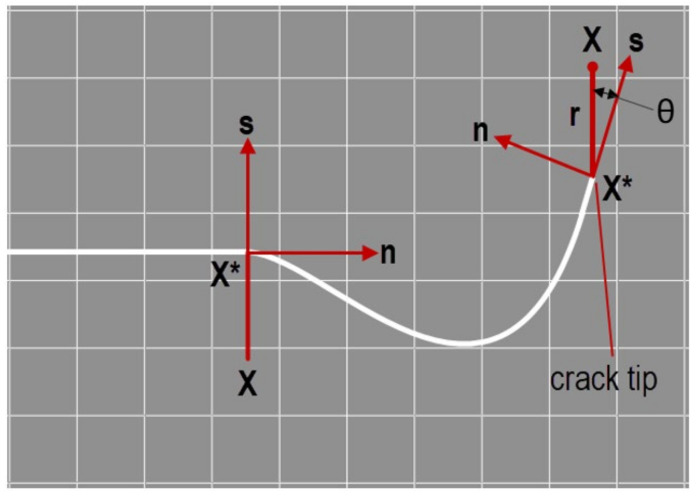
Normal and tangential vectors for a smooth fracture.

**Figure 5 materials-16-04714-f005:**
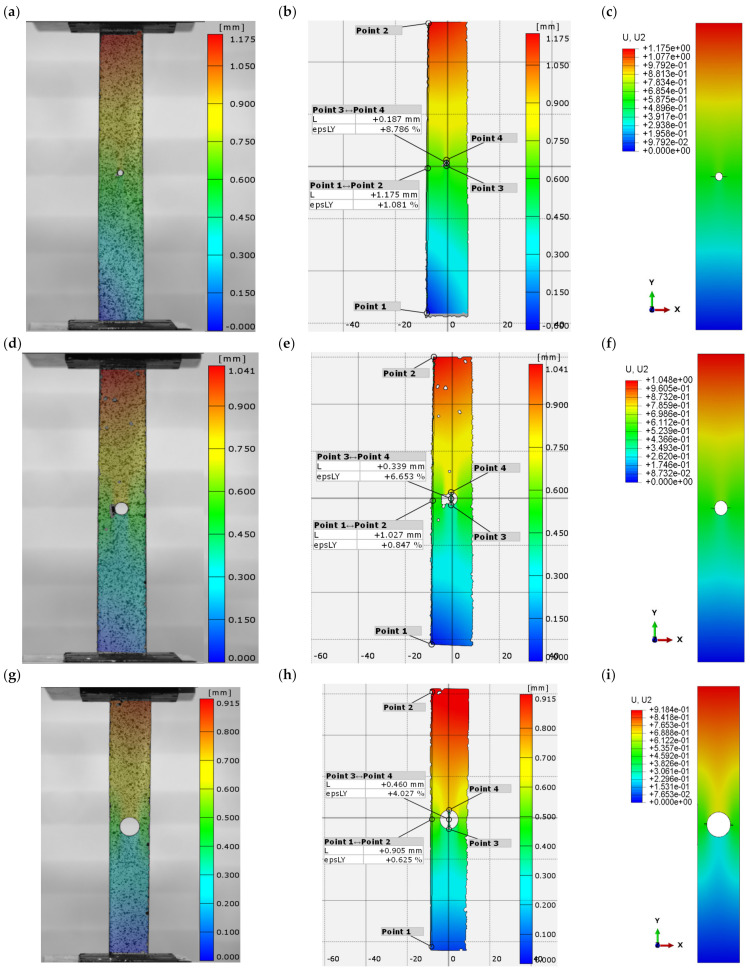
Hole elongation maps: (**a**,**b**) EXP_H_2 mm, (**c**) FEM_H_2 mm, (**d**,**e**) EXP_H_4 mm, (**f**) FEM_H_4 mm, (**g**,**h**) EXP_H_8 mm, (**i**) FEM_H_8 mm.

**Figure 6 materials-16-04714-f006:**
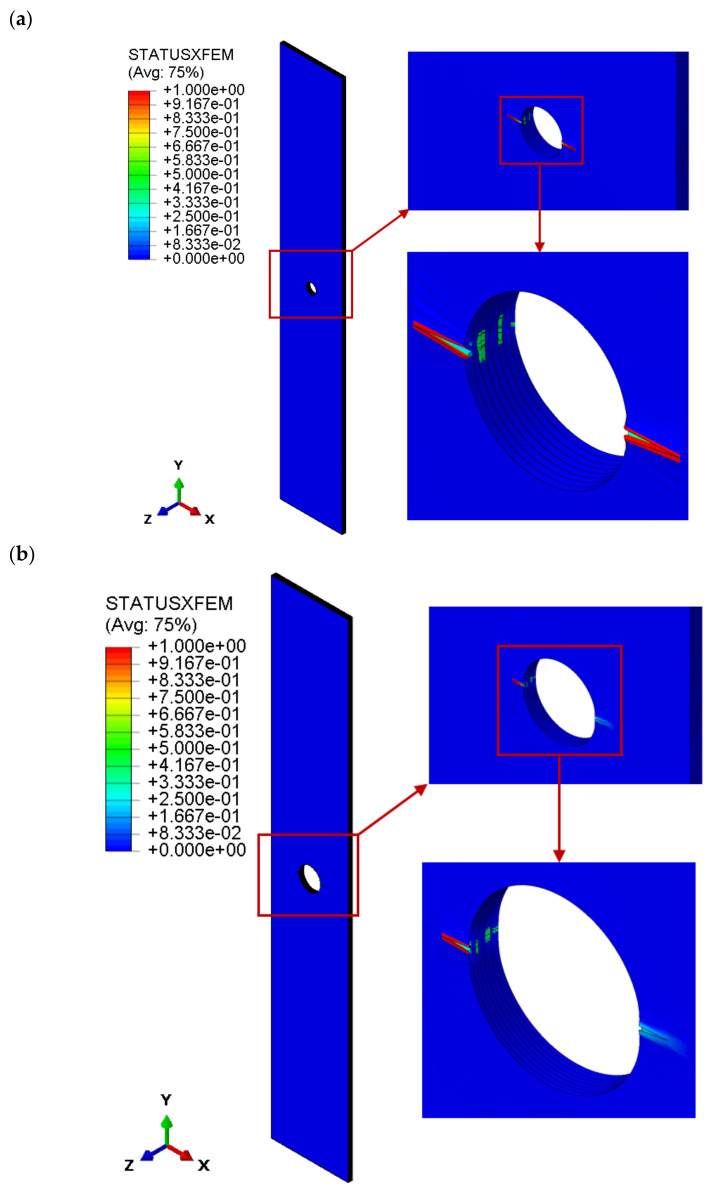
Damage and crack propagation in the composite structure: (**a**) FEM_H_2 mm, (**b**) FEM_H_4 mm, (**c**) FEM_H_8 mm.

**Figure 7 materials-16-04714-f007:**
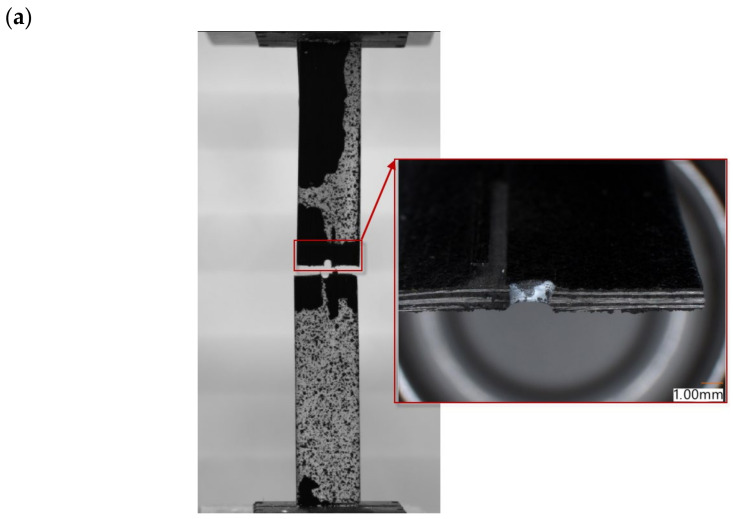
Complete failure of the laminate structure: (**a**) EXP_H_2 mm, (**b**) FEM_H_2 mm, (**c**) EXP_H_4 mm, (**d**) FEM_H_4 mm, (**e**) EXP_H_8 mm, (**f**) FEM_H_8 mm.

**Figure 8 materials-16-04714-f008:**
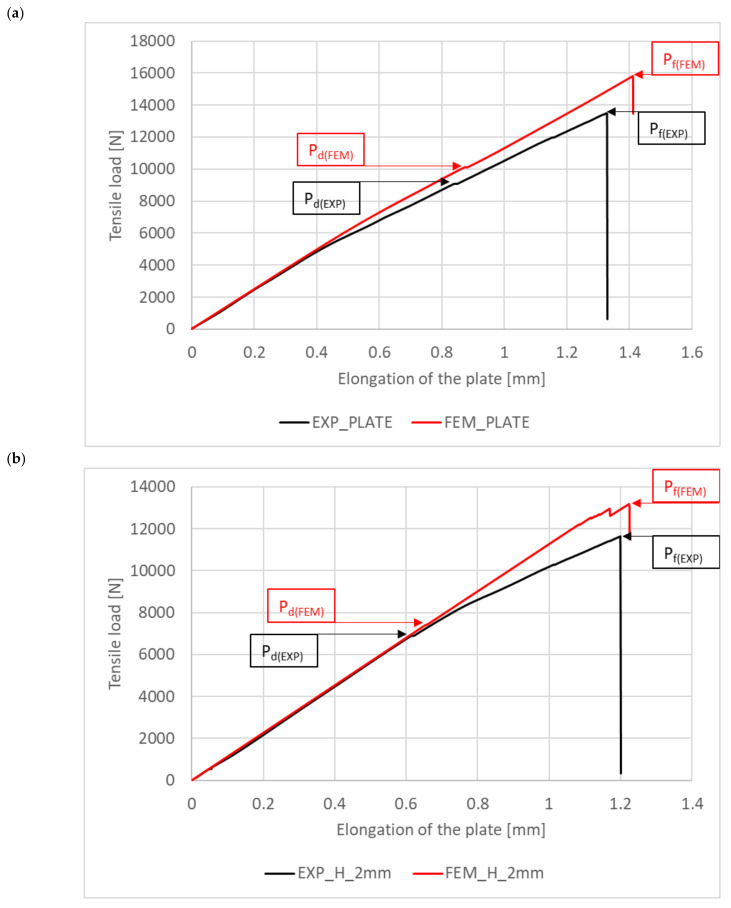
Work paths of the structure: (**a**) plate without a hole, (**b**) H_2 mm, (**c**) H_4 mm, (**d**) H_8 mm.

**Figure 9 materials-16-04714-f009:**
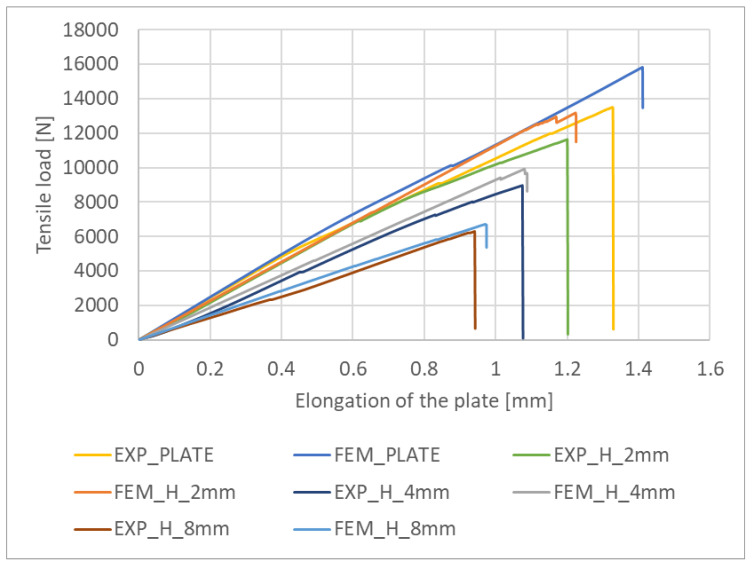
Comparison of experimental and numerical working paths.

**Table 1 materials-16-04714-t001:** Mechanical properties of CFRP.

Tensile Modulus		Shear Modulus	Poisson’s Ratio	Tensile Strength		Shear Strength	Compressive Strength	
E_1_	E_2,3_	G_12,13,23_	ν_12,13,23_	F_T1_	F_T2_	F_S_	F_C1_	F_C2_
GPa	MPa	MPa	–	MPa	MPa	MPa	MPa	MPa
130.71	6360	4180	0.32	1867.2	25.97	100.15	1531	214

**Table 2 materials-16-04714-t002:** Damage load and failure load of the composite structure.

	PLATE	H_2 mm	H_4 mm	H_8 mm
**P_d (EXP)_ [N]**	9031	6894	3938	2393
**P_d (FEM)_ [N]**	10,153	7430	4659	2801
**P_d_** _(error in prediction)_ **[%]**	11%	7%	15%	15%
**P_f (EXP)_ [N]**	13,484	11,635	8954	6299
**P_f (FEM)_ [N]**	15,672	13,179	9650	6683
**P_f_** _(error in prediction)_ **[%]**	14%	12%	7%	6%
**P_d (EXP)_ ÷ P_f (EXP)_↑ [%]**	49.31%	68.77%	127.37%	163.23%
**P_d (FEM)_ ÷ P_f (FEM)_↑ [%]**	54.36%	77.38%	107.13%	138.59%

## Data Availability

Not applicable.

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
