# Peer review of "Failure Mechanism of Tensile CFRP Composite Plates with Variable Hole Diameter"

_materials, 2023, doi:10.3390/ma16134714_

Round 1

Reviewer 1 Report

My comments are attached

English grammar has to be check again by authors

Reviewer 2 Report

Required extensive English editing.

Reviewer 3 Report

The review entitled "Effect of the circular hole on the cracking of the tensile composite plate” presents numerical and experimental results for composite plates with holes under axial tension. The main idea is to monitor the initiation and propagation of the cracks.

The author changed the main idea of the manuscript from crack propagation to a well-known topic, which is the effect of the hole diameter. Also, the author used an existing FE model. The manuscript lacks clarity, is weak in English, and cannot be accepted in the current format. This reviewer cannot find a new contribution to the state of the art.

Comments:

1-     The title should include the type of loading on this composite plate.

2-     Lines 18-21: The authors provided very weak conclusions. Providing a hole will cause weakness in the plate. The main findings of this manuscript should be highlighted in the abstract. However, this reviewer cannot find a novel conclusion to be addressed.

3-     The author should increase their discussion on previous related research and highlight how their study is providing a different approach or adding significantly to what has been done.

4-     Lines 80-81: The author mentioned “It is important to mention that the paper uses the currently very popular FEM method, which is used in many fields of engineering [49–57]”. Investigating the effect of holes in composite plates under static loading is not a new research topic as well as the author used an existing FE model. What are the new contributions that this manuscript will add to the state of the art?

5-     Figure 1: This reviewer recommends adding a sketch showing the dimensions of the specimen as well as the hole. All dimensions should be shown on this sketch.

6-     The developed numerical model is very weak. The author did not provide any information about the constitutive models of materials, damage modeling, and monitoring of the plastic zone around the initiated cracks. More discussions about the types of initiated cracks should be provided.

7-     The conclusion section should be shortened and concise. This reviewer suggests providing the conclusion section in bullets.

8-     The author needs to value the research work in this manuscript by conducting an extensive parametric study to explore the effect of different parameters on the behavior of composite plates with holes.

The Quality of the English Language is not good enough. Moderate editing of the English language is required.

Round 2

Reviewer 1 Report

The reviewer would like to thank the authors for their revision of the manuscript. I have a few comments, as follows:

1. the microscopic image presented in Figure 1 is blurred and has no scale. please modify

2. In the revised version, figure 4 should appear beside Figure 3, therefore,  please combine them.

3. In response to “Point 14”, the authors described the ARMIS System as “The ARAMIS optical-measuring system is designed for non-contact displacement measurements in planar and spatial components under load. It consists…” Please include a short explanation of it in the manuscript.

4. Formulation of displacement calculation through the ARAMIS system is still required for the standard presentation of the system used.

5. Point 17 (a,b,c,d);  not all the comments are responded to. The arrangement of the data to adequately explain the failure phenomena could be modified for an effective presentation of the results. For example, you could first explain the displacement field (figure 7) and then talk about cracking (figure 6). This is very important

6. The quality of some images needs modification, I am not sure why they are blurred. Example: figures 7 (legend).

English Grammar double-checking is required 

Reviewer 2 Report

The revised manuscript of this paper, with a full revision of the introduction and references, can be recommended for publication in this journal.

Moderate English editing is required.

Author Response

Thank you for reviewing my article. All comments are very important to me. Your comments have made my manuscript better 

Reviewer 3 Report

The author has addressed most of the reviewer's comments.

Author Response

(The authors gave the same response as above.)
